# Identifying actionable driver mutations in lung cancer using an efficient Asymmetric Transformer Decoder

**Biagio Brattoli\***                                           BIAGIO@LUNIT.IO

**Jack Shi\***                                                      JACK@LUNIT.IO
*Lunit Oncology*
*Germany*

**Jongchan Park**                                          JCPARK@LUNIT.IO

**Taebum Lee**                                       TAEBUM.LEE@LUNIT.IO

**Donggeun Yoo**                                          DGYOO@LUNIT.IO

**Sergio Pereira**                                         SERGIO@LUNIT.IO
*Lunit Oncology*
*Seoul, South Korea*

## Abstract

Identifying actionable driver mutations in non-small cell lung cancer (NSCLC) can impact treatment decisions and significantly improve patient outcomes. Despite guideline recommendations, broader adoption of genetic testing remains challenging due to limited availability and lengthy turnaround times. Machine Learning (ML) methods for Computational Pathology (CPath) offer a potential solution; however, research often focuses on only one or two common mutations, limiting the clinical value of these tools and the pool of patients who can benefit from them. This study evaluates various Multiple Instance Learning (MIL) techniques to detect six key actionable NSCLC driver mutations: ALK, BRAF, EGFR, ERBB2, KRAS, and MET ex14. Additionally, we introduce an Asymmetric Transformer Decoder model that employs queries and key-values of varying dimensions to maintain a low query dimensionality. This approach efficiently extracts information from patch embeddings and minimizes overfitting risks, proving highly adaptable to the MIL setting. Moreover, we present a method to directly utilize tissue type in the model, addressing a typical MIL limitation where either all regions or only some specific regions are analyzed, neglecting biological relevance. Our method outperforms top MIL models by an average of 3%, and over 4% when predicting rare mutations such as ERBB2 and BRAF, moving ML-based tests closer to being practical alternatives to standard genetic testing.

**Keywords:** Computational Pathology, Cancer Driver Mutation, Multiple Instance Learning, Deep learning.

## 1 Introduction

Advances in genomics along with personalized medicine have transformed the treatment of lung cancer, thereby enhancing the survival rates among patients Herbst et al. (2018). The improvement in patient outcomes was significantly influenced by the creation of targeted agents aimed at genetic mutations driving cancer growth Oudkerk et al. (2021), such as EGFR and KRAS . These mutations are considered *actionable* as the primary treatment

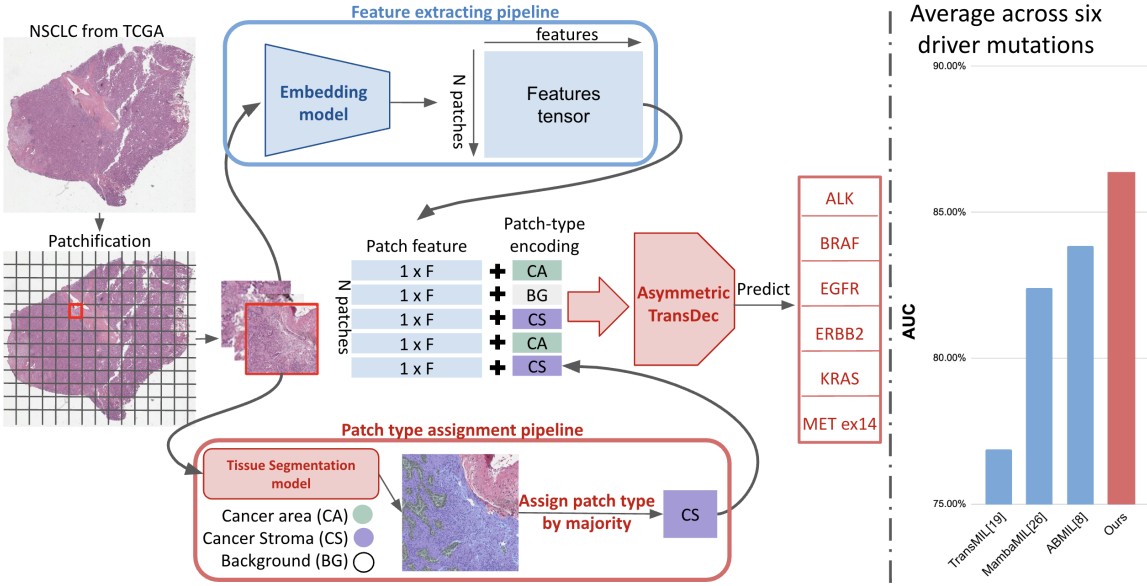

Figure 1: Left) Overview of our MIL approach. First, the WSI is patchified and features for each patch are extracted using an embedding model, typically an FM. Second, each patch is assigned a biologically meaningful tissue type using a segmentation model (stratification). For each tissue type, a learnable encoding is assigned and added to the respective patch. Finally, both features and tissue type are fed into our Asymmetric Transformer Decoder, which is trained to predict a genetic mutation. Right) Average AUROC over predicting 6 mutations for four MIL methods on our internal dataset.

recommendation is dependent on the presence of such mutations. Thus, performing genetic tests for precise identification of actionable mutations is essential for optimal treatment decisions and has been included in international guidelines Planchard et al. (2018). However, the adoption of next generation sequencing (NGS) testing encounters several obstacles including high costs, the need for specialized infrastructure, long turnaround times, and the necessity for sufficient tissue samples. These issues frequently result in patients receiving suboptimal treatment Habli et al. (2020). In contrast, histopathological analysis of H&E stained tissue is a standard process in cancer care and readily available within hospitals and other healthcare facilities. Therefore, ML-based tools for the identification of driver mutations in H&E-stained whole slide images (WSIs) would address some of the limitations of genetic testing. The ML model can function as a screening tool to identify those who are highly unlikely to have a positive driver mutation outcome, thus reducing expenses and unnecessary delays in standard treatment while awaiting a likely negative genetic test Shmatko et al. (2022).

Conventional ML methods are unsuitable for WSIs as their size exceeds GPU memory capacity. Consequently, Multiple Instance Learning (MIL) has emerged as the primary approach for computational pathology (CPath) tasks. Several MIL approaches have been investigated in CPath, including Graph Neural NetworksZheng et al. (2022); Shi et al.

(2024b), attention Ilse et al. (2018); Chen et al. (2024), transformers Shao et al. (2021); Wagner et al. (2023), vision-language Shi et al. (2024a), Mamba Yang et al. (2024), and others Zheng et al. (2024); Li et al. (2024). While there are several MIL alternatives, the existing work on the task of lung driver mutation prediction is relatively scarce Wagner et al. (2023), with the majority focusing on EGFR Campanella et al. (2022); Pao et al. (2023). The scarcity of studies on the automated identification of multiple actionable mutations reveals a notable research gap. In contrast, in this study, we address several clinically relevant lung cancer driver mutations.

ML can detect genetic mutations from histology images as driver mutations in cancer correlate strongly with specific histologic phenotypes and alter cell morphology and the tumor micro-environment Villa et al. (2014), visible in tissue samples. Nevertheless, many MIL studies ignore the semantic significance of each tissue patch and simply use all of them from the WSI. Some approaches select patches with cancer tissue to help the model capture more relevant features. Liu et al.Liu et al. (2024) show that focusing the model on particular WSI regions is viable by using tissue segmentation as attention mask labels. Our study corroborates the observation that incorporating biologically meaningful information into MIL is powerful, but, in contrast to Liu et al. (2024), we incorporate tissue semantics directly into the model input instead of using it as target.

In summary, we present three contributions. 1) First, a comprehensive benchmark of state-of-the-art CPath MIL methods for predicting clinically relevant NSCLC actionable mutations. 2) Second, a novel integration of tissue biology into transformer-based MIL through tissue-type encoding. 3) Finally, we introduce the Asymmetric Transformer Decoder, a novel architecture that significantly reduces model redundancy, enabling more efficient and accurate mutation prediction. Fig. 1 shows an overview of the proposed method.

Crucially, this work aligns with patient and clinical needs for effective targeted therapies within the diagnostic workflow by focusing on the prediction of actionable mutations in NSCLC from widely available H&E WSIs.

## 2 Method

Bag-wise MIL framework is the standard approach for WSI classification tasks because of the large size of the images. Typically, it involves the following steps: first, each WSI is divided into non-overlapping patches; second, a feature extractor is used to map each patch into a lower-dimensional embedding space; finally, an aggregator combines the patch embeddings and infers a prediction.

### 2.0.1 TRANSFORMER DECODER FOR EFFECTIVE INFORMATION EXTRACTION.

Foundation Models (FMs) are trained on extensive data and possess large model sizes, enabling them to produce powerful patch embeddings. The main challenge for an MIL aggregator is to distill pertinent data from a large number of patches into a cohesive representation while reducing overfitting to extraneous details such as source or scanner type that FM may also capture Yun et al. (2024). The transformer's multi-head attention mechanism is ideal for handling long sequences and theoretically well-suited for aggregation. Nevertheless, transformers struggle to meet expectations, still contending with simpler attention

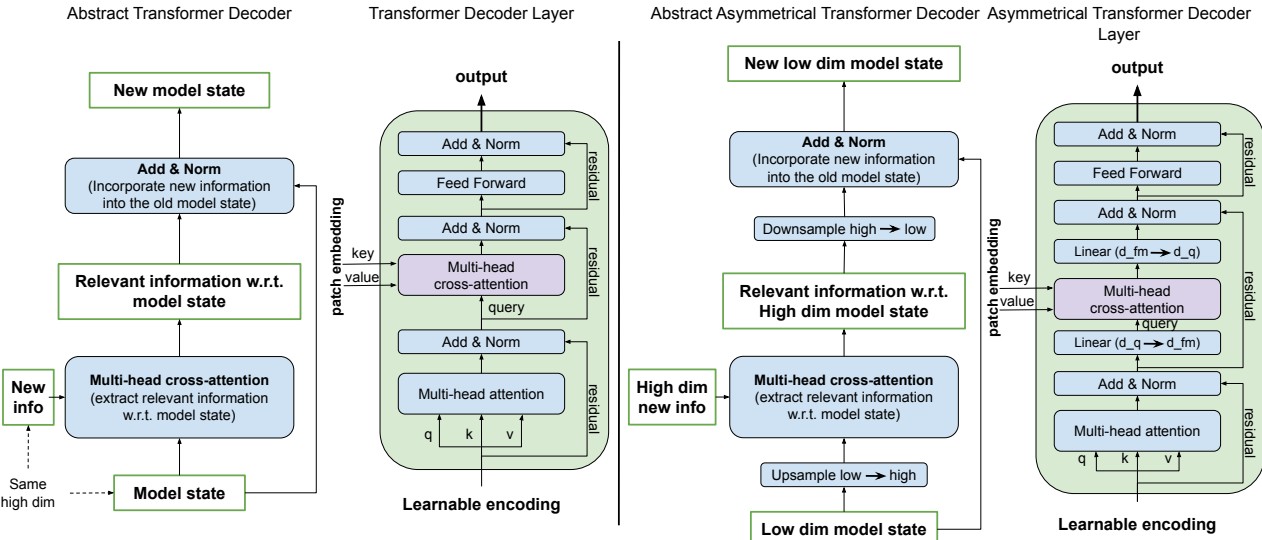

Figure 2: Left) The traditional Transformer Decoder setup for MIL. Right) Our Asym-TransDec leverages small learnable parameters in the initial attention module. We then project the query to align with patch embedding dimensions via a linear projection prior to cross attention. The output is reduced back to the small original dimension through a linear mapping before integrating into the residual connection.

models Chen et al. (2024). Despite the fact that FMs already encode patch details, transformer **encoders** are redundantly used to re-embed representations for downstream tasks Shao et al. (2021), resulting in inefficiency. Conversely, transformer **decoders** are tailored for pulling out essential features from lengthy encoded vector sequences. The transformer decoder uses a set of learned queries to extract data from the encoder output via cross attention without further encoding and transforming the learned patch features, as represented in Fig. 2-left. The standard scaled dot-product attention Vaswani et al. (2017) is given by the following:

$$Attention(Q, K, V) = \text{softmax}(\frac{QK^T}{\sqrt{d_k}})V$$

where $Q, K, V$ are respectively query, key and value, three learnable encoding vectors of size $d_q$ and $d_{kv}$. In cross attention, the dimensions of the query, key and value encodings must coincide, i.e. $d_q = d_{kv}$, usually aligning with the high dimensionality of FM embeddings. Increasing the size of query-key-value dimensions can significantly enhance a Transformer model's capacity; however, this may lead to overfitting. Projecting the FM embedding to a smaller dimension is a common approach to mitigate this risk, although it results in notable information loss. Hence, we opt to maintain full access to the $d_{kv}$ dimension embeddings to prevent such loss. Consequently, an asymmetric approach with different dimensions for query and key-value is necessary. Under these constraints, we create a model using a compact set of low-dimensional vectors $q$ as queries to lessen overfitting risks. To

Table 1: Evaluation of the proposed method with CNN- and ViT-based FMs, and comparison with SotA methods. Results are the average AUROC on 5-fold cross-validation.

| Embed | Aggregator | ALK | BRAF | EGFR | ERBB2 | KRAS | METex14 | *AVG* |
|-------|-----------|-----|------|------|-------|------|---------|-------|
| | #Samples Pos/Neg | 70/1130 | 100/1123 | 275/940 | 78/1145 | 184/1025 | 88/1135 | |
| CNN | ABMILIlse et al. (2018) | 83.7% | 80.1% | 85.1% | 75.7% | **76.5%** | 80.1% | 80.2% |
| | TransMILShao et al. (2021) | 81.2% | 78.1% | 83.6% | 70.9% | 74.6% | 77.5% | 77.6% |
| | MambaMILYang et al. (2024) | 82.0% | 80.5% | 85.5% | 72.9% | 74.0% | 79.8% | 79.1% |
| | Ours | **85.2%** | **82.8%** | **86.7%** | **78.7%** | 76.4% | **80.8%** | **81.8%** |
| ViT | ABMILIlse et al. (2018) | 87.5% | 82.4% | 89.3% | 75.5% | 84.2% | 84.1% | 83.8% |
| | TransMILShao et al. (2021) | 80.0% | 79.8% | 85.8% | 66.3% | 73.2% | 76.2% | 76.9% |
| | MambaMILYang et al. (2024) | 88.1% | 82.2% | 89.0% | 73.2% | 84.2% | 77.7% | 82.4% |
| | Ours | **88.4%** | **86.5%** | **90.8%** | **80.1%** | **85.9%** | **86.4%** | **86.3%** |

prevent losing information due to smaller patch embeddings, we transform the query tokens $q$ from a small dimension $d_q$ (e.g., 64) to a larger dimension $d_{kv}$ (e.g., $1,536$, as in ViT features) for cross attention alignment using a linear projection. The cross attention output is then downscaled to $d_q$ and integrated with the query through a residual connection. The detailed architecture is depicted in Fig. 2-right. The discrepancy in dimensions between query tokens and patch embeddings (keys, values) is the reason our approach is named Asymmetric Transformer Decoder. The asymmetric cross-attention is

$$AsymmetricAttention(Q_{dq}, K_{dkv}, V_{dkv}) = \text{softmax}(\frac{Q_{dq}W_{[dq,\ dkv]}K_{dkv}^T}{\sqrt{d_{kv}}})V_{dkv}W_{[dkv,\ dq]}$$

### 2.0.2 INTEGRATING BIOLOGICAL INFORMATION

Recent MIL methods have standardized identifying cancer regions within tissue samples Campanella et al. (2022). Basic image processing is first employed to remove irrelevant patches Campanella et al. (2022). Subsequently, a segmentation model Ronneberger et al. (2015); Mathis et al. (2018) is trained to isolate the cancer area for patch extraction. This method enriches the MIL input with diagnostically pertinent features; however, we hypothesize that focusing on cancer regions may discard other relevant information. Thus, rather than filtering patches, we suggest providing the model with biological information. We utilize a DeepLabV3-based Mathis et al. (2018) model to segment three tissue classes: cancer area (CA), cancer stroma (CS), and background (BG). Patches are labeled by the dominant class, defined as occupying over 50% of the patch. A unique learned encoding for each tissue type is added to the respective patch embeddings, as illustrated in Fig. 1. We refer to this process as **tissue-type stratification**. Importantly, this allows the model to adapt to each patch differently based on their innate biology, without discarding potentially useful

Table 2: Performance of actionable mutation prediction (AUROC) in the TCGA external test set. Models were trained on the complete internal dataset using ViT Saillard et al. (2024).

| Model | EGFR | KRAS | MET |
|---|---|---|---|
| ABMIL | 86.2% | 80.3% | 83.9% |
| TransMILShao et al. (2021) | 87.3% | 80.0% | 72.7% |
| MambaMILYang et al. (2024) | 86.3% | 82.2% | 81.2% |
| Ours | **88.1%** | **84.4%** | **87.5%** |

information available outside of CA. Stratification also helps the model to pay attention to each meaningful region, without being biased by the amount of each tissue type within the WSI. In fact, from each WSI, we randomly sample patches in the ratios of 50% cancer area, 30% cancer stroma, and 20% background.

## 3 Results

### 3.0.1 Dataset

The dataset comprises 1,223 surgical resection WSIs assessed via NGS. Among all mutations, we utilize six actionable mutations as labels: ALK, BRAF, EGFR, ERBB2, KRAS, and MET_ex14. Tab. 1 shows the number of available samples for each task. The WSIs originate from various hospitals across Western and Asian regions, digitized using Leica Aperio scanners (AT Turbo, AT2, GT450). An external test set consists of 924 NSCLC slides from the Cancer Genomic Atlas (TCGA) dataset Albertina (2016); Kirk (2016). Due to the scarcity of ALK, BRAF, and ERBB2 positives in TCGA, we focus our testing on EGFR, KRAS, and MET.

### 3.0.2 Implementation details and experimental setup

Existing FMs Kang et al. (2023); Saillard et al. (2024) trained on large unlabeled histopathology datasets served as feature extractors. The models were designed in PyTorch, extracting patch embeddings with 4 NVIDIA T4 GPUs, with size of $224 \times 224$ pixels at $20\times$ magnification. The embedding dimensions are $1,536$ for ViT and $768$ for CNN. For aggregator training, the AdamW optimizer with a learning rate of 0.0002 and a cosine annealing scheduler ($T = 10$ epochs, $\eta_{\min} = 1e - 6$) was used. The transformer decoder model utilized a batch size of 128 WSIs. In the transformer model, multi-head attention featured 2 heads, model dimension (query dimension $d_q$) of 64, GeLu activation, and dropout ($p = 0.5$) before the classifier layer.

The target mutations are rare; therefore, we develop the method following a stratified 5-fold cross-validation scheme using the internal dataset (1,223 WSIs). We further mitigate overfitting by tuning the hyper-parameters using only fold 0 from the ALK task and applying them across all tasks and folds. Each mutation is considered a separate task to further tackle class imbalance. In the end, we use the full 1,223 WSIs dataset for training and evaluate

Table 3: Effect of different tissue types on mutation task performance. Results are the average AUROC on 5-folds. The ViT-based FM Saillard et al. (2024) was used as patch embedding.

| Tissue type | ALK | BRAF | EGFR | ERBB2 | KRAS | METex14 | *AVG* |
|---|---|---|---|---|---|---|---|
| BG-only | 83.7% | 79.2% | 84.6% | 69.4% | 74.1% | 82.8% | 79.0% |
| CS-only | 87.4% | 84.9% | 88.4% | 77.3% | 82.7% | 80.8% | 83.6% |
| CA-only | 87.1% | 84.0% | 89.2% | 77.7% | 83.4% | 84.7% | 84.4% |
| All patches, no strat. | 88.1% | 85.5% | 90.4% | 78.6% | 85.0% | 85.5% | 85.5% |
| **All patches, stratified** | **88.4%** | **86.5%** | **90.8%** | **80.1%** | **85.9%** | **86.4%** | **86.3%** |

in the external test set. We use the area under the receiver operating characteristic curve (AUROC) as the evaluation metric.

### 3.0.3 DETECTING ACTIONABLE DRIVER MUTATION

Tab. 1 shows 5-fold cross-validation results across the six actionable mutations. We compare our results with three representative state-of-the-art aggregators: ABMIL Chen et al. (2024), TransMIL Shao et al. (2021), and MambaMIL Yang et al. (2024). We observe that the proposed method outperforms other aggregators in all mutations (except for KRAS with CNN-based FM). Notably, the gap is larger in relation to TransMIL (CNN FM: 4.2%; ViT FM: 9.4%), which is also based on Transformer. Indeed, TransMIL obtains the lowest metrics across the considered methods. This suggests that applying Transformers to MIL is not straightforward, requiring a careful formulation, such as the proposed method. We can also observe that TransMIL and MambaMIL achieve better results with the CNN-based FM, while the more lightweight ABMIL can leverage the powerful ViT-based features. In contrast, the proposed method achieves superior results with both FMs. We hypothesize that it is due to the efficient formulation of the proposed Asymmetric Transformer Decoder, which allows it to leverage the higher-dimensional ViT-based features.

We perform external validation of the proposed method on three mutations: EGFR, KRAS, and MET ex14. This is due to the lack of sufficient positive samples in the remaining mutations. Results are presented in Tab. 2. We observe a similar trend as before, where the proposed method obtains the top performance. Moreover, the performance level of our method is consistent, with 88.1% for EGFR, and 84.4% for KRAS. In contrast, TransMIL performs well for EGFR prediction (87.3%), but severely under-performs in MET (72.7%).

### 3.0.4 ABLATION STUDY

Our contribution includes integrating biologically meaningful tissue regions into the model. As shown in Tab. 3, we analyze the contributions of BG, CS, and CA tissues. The CA-only results lead, followed by CS-only and BG-only, in line with expectations. All regions show discriminative capability (see Tab. 1, BG-only outperforms TransMIL). This might partly result from segmentation errors in miss-assigning patches, and we also propose that cancer, and certain mutations, can alter adjacent tissues. Results without tissue-type stratification,

Table 4: Comparison between transformer architectures, using features from the ViT-based FM. TransEnc and TransDec represent the regular transformer encoder and decoder, respectively. Results are the average AUROC on 5-fold cross-validation.

| Tissue type | ALK | BRAF | EGFR | ERBB2 | KRAS | METex14 | *AVG* |
|---|---|---|---|---|---|---|---|
| TransEnc | 86.7% | 78.8% | 89.0% | 66.4% | 81.1% | 70.7% | 78.8% |
| TransDec | 88.3% | 82.7% | 89.9% | 79.2% | 84.5% | 85.0% | 84.9% |
| **AsymTransDec** | **88.4%** | **86.5%** | **90.8%** | **80.1%** | **85.9%** | **86.4%** | **86.3%** |

where all patches are equally considered as in standard MIL, outperform considering specific ROI but underperform compared to the proposed tissue-type stratification. This indicates that all WSI regions can aid mutation prediction, yet the stratification method enhances information extraction.

In Tab. 4, we compare the proposed Asymmetric Transformer Decoder aggregator with vanilla Transformer Encoder and Decoder aggregators. The encoder-based aggregator underperforms the decoder-based aggregators by a large margin. The encoder is a large-capacity model aimed at learning powerful representations. However, this leads to difficulties during training and overfitting in a MIL setting with few positive samples. Instead, the decoder is more lightweight and focused on integrating information from the already powerful FM-based features. We observe that even the vanilla decoder obtains better driver mutation prediction performance (84.9%) than the encoder (78.8%). The propose Asymmetric Transformer Decoder further boosts the performance to 86.3%, demonstrating that its efficient design is more adequate as an MIL aggregator.

## 4 Conclusion

Identifying driver mutations in lung cancer is essential for targeted therapies and improving patient prognoses. Yet, genetic testing is often slow, expensive, and requires specialized facilities, limiting its use and delaying treatment in many clinics. To overcome these barriers, we developed the Asymmetric Transformer Decoder model, an MIL-based technique that examines typical H&E-stained whole-slide images (WSIs) to identify mutations. Unlike conventional methods focusing only on cancer areas, our model includes tissue-type embeddings to further utilize data from cancer stroma and background regions, boosting mutation detection. Experiments reveal that our model surpasses leading MIL models, notably for rare mutations like ERBB2 and BRAF. This indicates its potential as a scalable, cost-effective genetic testing alternative suitable for local deployment in various clinical contexts, enhancing access to precision oncology.

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
