# OpenReview forum: "Identifying actionable driver mutations in lung cancer using an efficient Asymmetric Transformer Decoder"
_MICCAI.org/2025/Workshop/COMPAYL — COMPAYL 2025_

### Official Review · Reviewer_TEha · 2025-07-11
**Meaningful study, but clinically not convincing**

**Rating:** 3
**Confidence:** 5

**Review:**

This paper proposes an Asymmetric Transformer Decoder for predicting six actionable driver mutations in NSCLC from H\&E-stained WSIs using a MIL-based framework. The method includes tissue-type stratification and achieves strong results in internal validation, as well as external testing on TCGA for a subset of mutations. The technical contribution is sound and the inclusion of multiple mutation targets represents a step forward compared to prior EGFR-focused work.

Strengths:
* Addresses a clinically relevant task at the intersection of pathology and precision oncology.
* Introduces a novel transformer-based MIL aggregator that is efficient and robust to overfitting.
* Incorporates biologically meaningful tissue-type information into the model design.
* Broadens the focus beyond EGFR to include more mutations
* Results are benchmarked against state-of-the-art methods, with consistent performance gains.

Weaknesses:
* The list of mutations remains incomplete relative to clinical guidelines (e.g., ROS1, RET, NTRK), with no explanation for inclusion/exclusion criteria. Their cited ESMO guidelines [14] are heavily outdated.
* The authors highlight superior performance on ERBB2 (Her2) and BRAF, but these were not externally validated, which weakens the claim. Regarding this results, the authors overlook the fact that IHC is already available and widely used for detecting Her2 and BRAF, making the claim of a “cost-effective alternative” to molecular testing misleading.

Comment:
While the paper introduces meaningful technical innovations and expands the mutation prediction task beyond typical approaches, the clinical narrative is insufficiently supported. Key claims (especially regarding ERBB2/BRAF and model utility) are overstated, and the lack of justification for mutation selection limits clinical relevance. These concerns could be addressed in a careful revision, but as submitted, the paper falls short of being fully convincing.

---

### Official Review · Reviewer_uvqL · 2025-07-14
**MIL based approach for identifying six key mutations in NSLC**

**Rating:** 3
**Confidence:** 4

**Review:**

This paper presents an MIL based approach to detect six key mutations in NSCLC H&E WSIs. The authors present an approach using asymmetric transformer decoder which utilizes patch embeddings followed by tissue segmentation for making predictions. Three main contributions have been claimed by the authors.

1) Benchmark of state-of-the-art CPath MIL methods for predicting clinically relevant NSCLC mutations.
- I am not sure if this is a novel contribution as there are existing benchmarks for this e.g., Coudray et al. 2018 (https://www.nature.com/articles/s41591-018-0177-5)

2) A novel integration of tissue biology into transformer-based MIL through tissue-type encoding.
- This can be argued as several models segment tissue regions before applying ML approaches. May be not for "transformer-based MIL".

3) The Asymmetric Transformer Decoder,
- This is one of the key contributions of the paper.

The method has been compared using an internal 5-fold cross validation split and tested on TCGA data on some of the key mutations due to limited availability of data on others in TCGA. My main concern is that the authors have not compared against any publicly available benchmarks on NSCLC, e.g., Coudray et al. 2018, which makes the validation weak.

---

### Official Review · Reviewer_R8iG · 2025-07-21
**New Transformer Architecture for Mutation Prediction Incorporating Tissue Information**

**Rating:** 3
**Confidence:** 4

**Review:**

This paper introduces an Asymmetric Transformer Decoder architecture for predicting actionable driver mutations in NSCLC from histopathology slides. It benchmarks against state-of-the-art MIL methods and incorporates tissue-type stratification to improve biological relevance. The model achieves strong performance, particularly on rare mutations, but omits key implementation details.

Pros
1. The proposed Asymmetric Transformer Decoder reduces redundancy by using low-dimensional queries and higher-dimensional key-values, improving efficiency.
2. Tissue-type stratification via DeepLabV3 segmentation and patch-level encoding adds meaningful biological context to the MIL pipeline.
3. Outperforms ABMIL, TransMIL, and MambaMIL.
4. Uses a diverse dataset of 1,233 WSIs from multiple regions and an external TCGA test set (924 slides), showing generalisability.
5. The authors state that their code will be made publicly available upon acceptance. This would be of benefit to the community and should be ensured.

Cons
1. The paper refers to CNN and ViT-based encoders as “foundation models” but does not specify which ones are used. This is a significant omission.
2. The performance of the DeepLabV3 tissue segmentation model is not reported, despite its critical role in the pipeline.
3. Lack of clarity around feature extraction, tissue segmentation and encoder architecture limits reproducibility and interpretability.
4. The study focuses on predicting driver mutations in lung cancer; broader validation across other pathology domains would enhance impact.